# Rapid Production of Nanoscale Liposomes Using a 3D-Printed Reactor-In-A-Centrifuge: Formulation, Characterisation, and Super-Resolution Imaging

**DOI:** 10.3390/mi14091763

**Published:** 2023-09-12

**Authors:** Yongqing He, Davide De Grandi, Stanley Chandradoss, Gareth LuTheryn, Gianluca Cidonio, Ricardo Nunes Bastos, Valerio Pereno, Dario Carugo

**Affiliations:** 1Department of Pharmaceutics, School of Pharmacy, University College London, London WC1N 1AX, UK; yongqing.he.20@ucl.ac.uk; 2Institute of Biomedical Engineering (IBME), Department of Engineering Science, University of Oxford, Parks Road, Oxford OX1 3PJ, UK; davide.degrandi@eng.ox.ac.uk; 3Oxford Nanoimaging Limited (ONI), Oxford OX2 8TA, UK; stan@oni.bio (S.C.); ricardo@oni.bio (R.N.B.); vpereno@oni.bio (V.P.); 4Nuffield Department of Orthopaedics, Rheumatology and Musculoskeletal Sciences (NDORMS), The Botnar Research Centre, University of Oxford, Windmill Road, Oxford OX3 7HE, UK; gareth.lutheryn@ndorms.ox.ac.uk; 53D Microfluidic Biofabrication Laboratory, Center for Life Nano- & Neuro-Science—CLN2S, Italian Institute of Technology (IIT), 00161 Rome, Italy; gianluca.cidonio@iit.it

**Keywords:** liposome, solvent exchange, centrifugal flow, microfluidics, super-resolution microscopy

## Abstract

Nanoscale liposomes have been extensively researched and employed clinically for the delivery of biologically active compounds, including chemotherapy drugs and vaccines, offering improved pharmacokinetic behaviour and therapeutic outcomes. Traditional laboratory-scale production methods often suffer from limited control over liposome properties (e.g., size and lamellarity) and rely on laborious multistep procedures, which may limit pre-clinical research developments and innovation in this area. The widespread adoption of alternative, more controllable microfluidic-based methods is often hindered by complexities and costs associated with device manufacturing and operation, as well as the short device lifetime and the relatively low liposome production rates in some cases. In this study, we demonstrated the production of liposomes comprising therapeutically relevant lipid formulations, using a cost-effective 3D-printed reactor-in-a-centrifuge (RIAC) device. By adjusting formulation- and production-related parameters, including the concentration of polyethylene glycol (PEG), temperature, centrifugation time and speed, and lipid concentration, the mean size of the produced liposomes could be tuned in the range of 140 to 200 nm. By combining selected experimental parameters, the method was capable of producing liposomes with a therapeutically relevant mean size of ~174 nm with narrow size distribution (polydispersity index, PDI ~0.1) at a production rate of >8 mg/min. The flow-through method proposed in this study has potential to become an effective and versatile laboratory-scale approach to simplify the synthesis of therapeutic liposomal formulations.

## 1. Introduction

Liposomes are spherical vesicular systems consisting of an aqueous core enclosed by one or more phospholipid bilayers [1]. Lipid-based nanoparticulate systems, such as liposomes and lipid nanoparticles, have gained enhanced momentum due to their applicability in the delivery of biologically active compounds, especially during the development of COVID-19 vaccines [2]. The amphiphilic nature of phospholipids, which comprise both hydrophilic and lipophilic moieties, make liposomes a suitable carrier for both hydrophobic and hydrophilic drugs, including biomacromolecules such as nucleic acids and proteins. Therefore, liposomes have the potential to improve bioavailability and reduce the toxic side-effects of a range of bioactive compounds, which can help in overcoming barriers to the clinical translation of novel classes of active pharmaceutical ingredients [3,4,5].

Several challenges, however, may hinder laboratory-scale development of liposomal formulations, including those associated with the complexity, high cost, and limited scalability of associated manufacturing processes. Batch methods are most commonly used to manufacture liposomes at the laboratory scale and pre-clinically, via techniques such as thin-film hydration, freeze–thawing, ethanol injection, and reverse-phase evaporation [6]. These manufacturing approaches are effective in producing liposomes of various compositions and have been widely used to manufacture drug-loaded formulations, including pH-sensitive [7,8], heat-sensitive [9], and theranostic [10,11] liposomes. However, due to the lack of accurate control over the physicochemical environment in batch production processes, multilamellar vesicles (or MLVs) with a broad size distribution are typically produced, requiring subsequent down-sizing procedures to obtain homogeneous, single unilamellar vesicles (SUVs) that are desirable for drug delivery applications [12]. These post-production procedures (such as extrusion and sonication) are effective at reducing the average size and narrowing the size distribution of liposomal dispersions [13]; however, their utility is limited by their time-consuming nature and the risk of sample contamination [14].

More recently, the development of microfluidic reactors has led to improved controllability over liposome production [15], as the transport of fluids and chemical species within micrometre-scaled channels is highly predictable and less difficult to characterise and manipulate, due to the associated laminar flow conditions [16]. Using microfluidic techniques, the production of SUVs with relatively uniform size can be achieved without the need for laborious post-production processes to fulfil requirements for pre-clinical and clinical application [17]. By changing the lipid formulation and the operating parameters, such as temperature and flow boundary conditions, the liposome properties can be tailored towards a specific application [18,19,20]. However, despite the benefits offered by microfluidic systems, increasing the lipid concentration and operating flow rates above certain limits can also result in reduced controllability over the obtained liposome properties and greater size dispersity. Therefore, the overall liposome production rate of custom-developed, laboratory-scale microfluidic methods is often relatively low (typically up to a few mg/min) [21]. Parallelisation of multiple microfluidic devices can improve production rates; however, time-consuming device fabrication procedures and the need for costly and specialist flow control units (i.e., syringe pumps and flow/pressure sensors) still limit the widespread adoption of microfluidic systems for liposome production [21]. Highly controlled environmental conditions are also required to prevent exogenous impurities from clogging the microfluidic channels; when used in a conventional laboratory setting, these devices thus often present a relatively limited lifetime. It should, however, be noted that different commercial systems have been developed in recent years that address some of the limitations of microfluidic techniques, demonstrating significantly greater liposome production rates (of up to few hundreds of mg/min) [22]. These systems, however, may not always be affordable for research laboratories, especially those within non-specialist and/or low-resource settings. Cost-effective production of microfluidic reactors has now been made possible by the burgeoning of 3D printing [23], with commercial benchtop 3D printers capable of manufacturing structures with dimensions of few micrometres [24]. In this research, we specifically utilised a 3D-printed reactor-in-a-centrifuge (or RIAC) device, previously developed by Cristaldi et al. [25], to evaluate the cost-effective, rapid, and facile production of liposomal formulations with constituents that are comparable to those employed in therapeutic formulations.

The formation of liposomes in the RIAC relies on the so-called ‘solvent exchange’ mechanism. In this method, amphiphilic phospholipid molecules are exposed to an increasingly hydrophilic environment, which induces their self-assembly into bilayer phospholipid fragments (BPFs) that subsequently form spherical vesicular systems [26]. The formation of liposomes via solvent exchange is therefore mainly governed by the mixing between an organic solvent (in which lipids are initially solubilised) and an anti-solvent (typically water or a saline solution). In conventional microfluidic-based methods, mixing between the solvent and anti-solvent can be achieved either via diffusion or advection (or a combination of these), with common device architectures including flow-focusing or serpentine-shaped channels [27,28]. Ethanol injection is regarded as a commonly used batch counterpart for these microfluidic techniques, whereby the lipidic organic solution is injected into a reservoir containing the anti-solvent [29] (Figure 1A). In the RIAC, the flows of solvent and anti-solvent are instead driven by a conventional laboratory centrifuge (Figure 1B), and the device has a cylindrical shape containing two reservoirs and a spiral-shaped mixing channel [25]. The lipidic solution and water are separately added to the reservoirs and, upon actuation of the centrifuge, they are driven through the spiral channel where they rapidly mix with each other, leading to lipid self-assembly into liposomes.

In previous research, we showed that the RIAC is capable of producing liposomes (as well as inorganic nanoparticles and drug nanocrystals) at a relatively low cost and without requiring specialist instrumentation [25,30]. These earlier proof-of-concept studies, however, employed model lipids that are less commonly used clinically for therapeutic purposes [31,32]. This also applies to several other studies using more conventional microfluidic devices, where the liposomal formulations often comprise model lipids with a transition temperature (*T_c_*) below room temperature [19,33]. This results in a simplification of the experimental conditions required for liposome formation. Importantly, the transition temperature of a lipid (i.e., the temperature at which the lipid transitions from the ordered gel phase to the disordered liquid crystalline phase) determines the rigidity of a lipid bilayer and can impact the physical properties and stability of the formed liposomes [34]. Moreover, liposomal formulations investigated in previous research often include only one or two constituents, whilst the number of excipients is typically greater in therapeutic formulations used clinically [35].

Considering these limitations, in the present study, we assessed the ability of the RIAC to produce a more clinically relevant formulation compared to the ones employed in previous research using this device [25]. The formulation was adapted from those used in some mRNA vaccines, as well as in other liposomal drug delivery products such as DaunoXome^®^, Onivyde^®^, and Vyxeos^®^ [36,37]. The liposome constituents used herein include 1,2-distearoyl-sn-glycero-3-phosphocholine (DSPC), cholesterol (chol), and a poly(ethylene glycol) (PEG)-containing moiety. DSPC has a relatively high transition temperature of 55 °C, and its incorporation within liposomes can result in improved stability and drug retention [38]. Cholesterol is known to modulate the fluidity and phase behaviour of lipid bilayers and has been reported to increase the packing of lipid molecules in bilayers as well as improve liposome resistance to aggregation [39]. Finally, PEG has been shown to improve liposome stability both in vitro and in vivo. To progressively improve the production protocol and achieve therapeutically desirable liposome properties, the following experimental conditions were varied: the presence of PEG, anti-solvent temperature, and centrifugation parameters. Liposome dimensions were quantified using dynamic light scattering (DLS) and, crucially, for the first time, super-resolution microscopy was employed to further investigate the dimensional characteristics of liposomes and provide evidence of their vesicular nature. To explore the versatility and capability of the RIAC in handling more complex liposomal formulations, the production of both cationic and biotinylated ligand-targeting formulations was also investigated. The former has been extensively used in the delivery of gene therapeutics (including in vaccines), while the latter has shown prospects in targeted delivery through both intravenous and peripheral administration routes [40,41,42].

## 2. Materials and Methods

An overview of all liposomal formulations investigated in the present study is reported in Table 1, including the corresponding method of liposome production. Liposome constituents comprised 1,2-distearoyl-sn-glycero-3-phosphocholine (DSPC), cholesterol (chol), polyoxyethylene (40) stearate (PEG40s), 1,2-dilauroyl-sn-glycero-3-ethylphosphocholine (chloride salt) (EPC), and 1,2-distearoyl-sn-glycero-3-phosphoethanolamine-N-[biotinyl(polyethylene glycol)-2000] (ammonium salt) (DSPE-PEG2000-biotin), which were all purchased from Sigma-Aldrich (Gillingham, UK).

### 2.1. Liposome Production via Ethanol Injection

Ethanol injection (controlled with a syringe pump) was chosen as a batch method in this study, as it represents a suitable macroscale counterpart for microfluidic methods that rely on the solvent exchange mechanism to produce liposomes [18,26]. The production protocol was adapted from the one used in a study by Carugo et al. [26]. Specific volumes of DSPC and cholesterol solutions in ethanol were transferred into a plastic syringe (Medicina, Bolton, UK). The syringe was then loaded onto a syringe pump (LEGATO^®^ 100, KD Scientific, Holliston, MA, USA), which was employed to convey the lipid solution into a water-containing vial at a flow rate of 500 μL/min. The vial was placed on a hotplate stirrer (Fisherbrand™ Isotemp™, Thermo Fisher Scientific, Loughborough, UK) set at 600 rpm and at varying temperature values to evaluate the effect of temperature on liposome properties. Once all the lipidic organic solution was injected into the vial, the mixture was stirred for a further 5 min. The obtained liposomal dispersion was stored at 4 °C until further characterisation.

### 2.2. 3D-Printed Reactor-In-A-Centrifuge (RIAC)

The RIAC was designed and manufactured through 3D printing, as reported previously. The technical drawings and printing specifications are freely available in a previous publication [25] to enable adoption (as well as design modifications) across different laboratories. The device was fabricated in polylactic acid (PLA) via fused deposition modelling (FDM) 3D printing using an Ultimaker S5 printer (Ultimaker, Utrecht, The Netherlands). Upon manufacturing, it was inserted into a conventional centrifuge tube and actuated by centrifugation. An initial group of tests was performed by adding water in both reservoirs and looking for potential fluid leakages through the outlet, in the absence of centrifugation. No leakage was detected in these experiments, indicating an effective liquid-retaining capacity. Liquid retention was provided by FRIT filters (pore size: 0.5 μm) placed at the bottom of each RIAC’s reservoir, as described previously [25]. To determine the minimum centrifugal force needed to process the full volume of solvent and anti-solvent, the device was further tested with 2 mL of water in each reservoir and 6 mL at the base of the centrifuge tube and was operated at different centrifugal forces, ranging from 500 to 2000 rcf (with centrifugation times in the range 1 to 10 min), using a 4-16KS refrigerated centrifuge (Sigma, Welwyn Garden City, UK).

### 2.3. Liposome Production Using the RIAC

For the production of liposomes and optimisation of the formulation parameters, 2 mL of the lipid solution (containing varying molar ratios of DSPC, cholesterol, and PEG40s) was pipetted into one reservoir. DI water was pre-heated to the desired temperature and pipetted into the other reservoir. Moreover, 6 mL of water was added at the base of a 50 mL centrifuge tube (SARSTEDT AG & Co., Nümbrecht, Germany) in which the RIAC was placed (as shown in Figure 1). The addition of water in the centrifuge tube was shown in previous research to reduce aggregation of the produced liposomes. Increasing this volume above 6 mL would not result in significant changes in liposome properties, but would further reduce the liposome concentration in the end-product [25]. The effects of varying the production parameters, including centrifugal force (in the range 1000–2000 rcf) and time (between 3 and 10 min), anti-solvent temperature (20 °C and 60 °C, corresponding to room temperature and a temperature above the *T_c_* of DSPC, respectively), as well as lipid composition and total lipid concentration (ranging from 2 mM to 20 mM), were investigated by measuring the size and size dispersity of the produced liposomes. The experimental conditions that individually led to the most favourable liposome properties were then combined to produce cationic (DSPC:Chol:EPC:PEG40s) and targetable (DSPC:Chol:DSPE-PEG-biotin:PEG40s) liposomes through the addition of EPC and DSPE-PEG2000-biotin as liposome constituents, respectively [43,44].

### 2.4. Measurement of Liposome Size and Charge through a Dynamic Light Scattering Apparatus

Dynamic light scattering (DLS) measurements were performed using the Zetasizer nano ZS instrument (Malvern Instrument Ltd., Malvern, UK) to determine the liposome size, size dispersity, and zeta potential (i.e., a measure of electrostatic charge). Three measurements were performed for each sample (refractive index = 1.33, absorption coefficient = 0, measurement angle = 173° backscatter). All measurements were performed at 25 °C after 120 s of equilibration time. The zeta-average liposome size, polydispersity index (PDI), and zeta potential were obtained from the Zetasizer software (Malvern Instrument Ltd., UK, version 7.11). The nanoparticle concentration was also measured in several specific experiments using the Zetasizer Ultra (Malvern Panalytical Inc., Malvern, UK). The viscosity values used for DLS measurements were calculated using the Zetasizer software, considering the relative amounts of solvent and anti-solvent in each sample.

### 2.5. Super-Resolution Imaging of Liposomes

Biotinylated liposomes were first labelled with a lipophilic photo-switchable dye using a proprietary protocol developed and optimised by ONI (Oxford Nanoimaging Ltd., Oxford, UK). In brief, the 20× diluted biotinylated liposomes were labelled with 10×-concentrated membrane stain. The liposomes were gently mixed with the membrane stain by pipetting the solution up and down. As reported in Figure 2A, the samples were then incubated at 4 °C overnight. Liposomes were immobilised and imaged inside a passivated and biotinylated imaging chip from ONI (Figure 2B). The imaging chip was brought to room temperature, taken out of its protective package, and stored in a clean and humid environment before loading the sample. Figure 2C briefly depicts the methodological procedure. An amount of 10 µL of neutravidin solution at 0.5 mg/mL concentration was added to each lane of the chip and incubated for 10 min. The unbound neutravidin was removed by washing the lanes with 2× 200 µL of 1× PBS solution.

An amount of 10 µL solution of labelled liposomes was then added to the lanes and incubated for 10 min. The unbound liposomes and the excess dye were washed with 2× 200 µL of 1× PBS. Prior to imaging, fresh imaging buffer was prepared by mixing 1 µL of Part A with 99 µL of Part B of ONI’s B3 Stochastic Optical Reconstruction Microscopy (dSTORM) buffer (Oxford Nanoimaging Ltd., UK). The imaging buffer was then introduced to the lanes containing captured liposomes. The liposomes were imaged using an ONI Nanoimager S equipped with a 640 nm laser at 85% laser power and a 30 ms exposure time. Each acquisition was performed for at least 3000 frames. The super-resolution images were further processed and analysed using ONI’s Collaborative Discovery platform CODI (www.alto.codi.bio) to obtain liposome size and morphology data.

### 2.6. Statistical Analysis

All statistical analyses were performed using GraphPad Prism 9 for macOS (GraphPad Software, LLC., Boston, MA, USA, version 9.0.1). For experiments conducted to optimize the liposome formulation and production protocol, three independent repeats were carried out. The results are shown as mean ± standard deviation of the mean (SD). One-way ANOVA was carried out to compare the size and PDI between different groups when different parameters were varied. Unpaired *t*-tests were used to specifically compare the ethanol injection method with the RIAC method, as well as to evaluate the effect of adding a PEG moiety and changing the anti-solvent temperature. For all analyses, a *p* value lower than 0.05 was considered as corresponding to a statistically significant difference between groups.

## 3. Results and Discussion

### 3.1. Effect of Temperature and Ethanol Fraction on Liposomes Produced via the Ethanol Injection (EI) Method

Several studies have demonstrated that cholesterol and DSPC tend to form stable liquid-ordered bilayer structures at molar ratios of up to 1:1 [45]. Therefore, in this study, a liposomal formulation comprising DSPC:chol (10:10 mM) was first evaluated, consistently with previous research [46]. Ethanol injection was chosen as a batch method, since it relies on the solvent exchange mechanism (as for the RIAC). Parameters including the temperature of the anti-solvent and ethanol fraction were initially varied to identify a suitable protocol for subsequent liposome production using the RIAC. Figure 3A shows the average liposome diameter and size dispersity (quantified as PDI) of liposomes produced at varying temperatures. A significant tendency (*p* value < 0.0001) of liposome size to reduce with increasing temperature was observed. When the anti-solvent temperature was increased to 60 °C, liposomes with a mean diameter of 174 ± 1.57 nm and low size dispersity (PDI = 0.17 ± 0.04) were produced, while at 20 °C, larger liposomes with a wider size distribution (size = 349 ± 30.5 nm, PDI = 0.66 ± 0.05) were produced. Therefore, the temperature of the anti-solvent (aqueous) phase had a significant effect on the final liposome dimensional characteristics, which is consistent with previous studies [39]. Notably, when the temperature was increased above the transition temperature of DSPC (*T_c_* = 55 °C), the reduction in liposome size and size dispersity could be attributed to reduced interdigitation in the lipid bilayer, which has been previously associated with liposome aggregation and fusion [47]. Decreased liquid viscosity at greater temperatures might also result in more efficient and faster mixing between solvent and anti-solvent, which has been associated previously with the formation of smaller liposomes [29,48].

The effect of varying the relative amount of ethanol was also investigated while keeping the total lipid concentration of the sample constant (Figure 3B). Liposomes produced at all three ethanol fractions evaluated in this study showed relatively low size dispersity (PDI < 0.2). Moreover, increasing the ethanol fraction appeared to decrease the PDI from 0.17 to 0.08. An ethanol fraction of 20% (*v*/*v*) resulted in liposomes of a smaller size (149 ± 1.94 nm) compared to an ethanol fraction of 10% (174 ± 1.57 nm) or 50% (182 ± 0.63 nm). The effect of varying the ethanol fraction on liposomes produced via ethanol injection has been well documented. Philippe et al. [49] evaluated its effect on liposome diameter, demonstrating that an increase in the ethanol fraction from 13% to 20% caused a decrease in liposome size, while an increase from 20% to 60% caused the liposome diameter to increase from 61 nm to 303 nm. Yoshie and colleagues [50] also demonstrated that an increase in ethanol fraction from 20% to 50% resulted in an increase in liposome size from approximately 200 nm to over 1000 nm. The effects of changing the amount of ethanol on liposome properties can be difficult to fully elucidate. At the lowest ethanol fraction used in this study (10%), a highly concentrated lipid solution was injected into the anti-solvent, and therefore a greater number density of lipid molecules was available locally during self-assembly to form larger liposomes [51]. However, increasing the ethanol fraction increased the time required to achieve complete mixing between solvent and anti-solvent. This, together with the reported ability of ethanol to induce fusion between liposomes, could result in an overall increase in liposome size [29], which would support the observed increase in liposome diameter when an ethanol fraction of 50% was employed.

### 3.2. Liposome Production Using the RIAC: Identification of Suitable Experimental Parameters

#### 3.2.1. Effect of Temperature

Since liposomes produced via ethanol injection had the lowest size and size dispersity when a 20% ethanol fraction and an anti-solvent temperature of 60 °C were employed, the liposome production protocol of RIAC was designed as follows. Amounts of 2 mL of lipid solution and 2 mL of water were separately pipetted into each reservoir, and 6 mL of water was added into the centrifuge tube (to achieve a final ethanol fraction of 20%). The dimensional properties of liposomes produced using the RIAC (‘RIAC liposome’) compared to those produced via ethanol injection (‘EI liposome’) are illustrated in Figure 4. At room temperature (RT) (20 °C), significant reductions in liposome size from 349 ± 30.5 nm to 171 ± 8.55 nm (*p* value < 0.0001) and in PDI from 0.66 ± 0.05 to 0.41 ± 0.11 (*p* value < 0.0001) were observed when the RIAC was employed as an alternative to batch ethanol injection. This could be due to the enhanced and faster mixing between solvent and anti-solvent in the spiral-shaped channels of the RIAC. When the anti-solvent temperature was increased to 60 °C, the size and PDI of liposomes produced by these two methods were comparable. In both methods, liposomes produced at 60 °C had a smaller size dispersity. This could be attributed to the role of temperature in lipid interdigitation and bilayer elasticity, as reported in previous work [52] and discussed above. The effect of temperature on liposome properties was not investigated in previous studies using the RIAC, where liposomes were only produced at room temperature. In these earlier studies, soybean phosphatidylcholine was employed as the main liposome constituent, being characterised by a low *T_c_* in the range of −30 to −20 °C. Therefore, liposomes with relatively small size (between 80 and 300 nm) and PDI (0.26) could be produced at room temperature. Results from the present study demonstrate that the RIAC can also be employed to rapidly produce liposomes from phospholipids with a greater transition temperature, such as DSPC, which are more commonly employed in clinical formulations [53]. Interestingly, the RIAC was capable of producing liposomes of comparable size when the anti-solvent temperature was both below and above the transition temperature of DSPC, likely due to the more vigorous mixing taking place within the reactor when compared to the ethanol injection process. The liposome size dispersity was, however, significantly reduced when the anti-solvent temperature was increased from 20 °C to 60 °C. It should be noted that pre-heating the anti-solvent phase does not provide a means of accurately controlling the liposome production temperature. Future work will specifically investigate novel RIAC design configurations that would enable more precise control over the medium’s temperature during liposome formation.

#### 3.2.2. Effect of Inclusion of a PEG Moiety

In order to further improve the end-product quality and its relevance to clinical formulations, a PEG moiety (i.e., PEG40s) was introduced in the liposomal formulation. PEG is known to increase the steric stability of particulate systems both in vitro and in vivo, due to the stealth effect provided by both its hydrophilicity and brush conformation [54]. Notably, some of the vaccines against COVID-19 are in the form of PEGylated lipid nanoparticles [55]. It was herein hypothesised that PEG would inhibit particle aggregation, thus resulting in liposome samples with a narrower size distribution. As shown in Figure 5A, when PEG40s was added at a concentration of 0.1 mM, a significant size reduction from 171 nm to 141 nm (*p* value < 0.0001) was observed at room temperature. This reduction in size could be attributed to the reduced tendency of lipid bilayer fragments to aggregate and form larger liposomes [56]. At a temperature of 60 °C, the addition of PEG40s resulted in only a slight, and not statistically significant, decrease in liposome size. The results reported in Figure 5B also show that there was a considerable effect of PEG on the size dispersity of liposomes produced with the RIAC, corresponding to PDI reductions from 0.41 to 0.24 (at 20 °C) and from 0.22 to 0.17 (at 60 °C), with *p* values < 0.0001. Overall, upon the addition of a PEG moiety, liposomes with therapeutically relevant size (172 ± 2.5 nm) and low size dispersity (PDI = 0.17 ± 0.02) could be produced at 60 °C.

These liposome samples were subsequently stored for one month to assess their stability. As shown in Figure 5C, the liposome formulation that did not contain PEG underwent an increase in both size and the PDI, whilst no significant change was observed for liposomes containing PEG, indicating the improved storage stability of the latter formulation. Due to the superior dimensional properties and stability of liposomes produced at 60 °C and containing PEG40s, the DSPC:Chol:PEG40s (10:10:1) formulation produced at an anti-solvent temperature of 60 °C was selected for further evaluation in subsequent experiments. It should be noted that an ethanol volume fraction of 20% did not appear to affect liposome stability upon storage (for the PEG-containing formulation); however, future work will focus on evaluating methods of removing residual organic solvent from the liposomal dispersion, as the presence of ethanol may affect the drug loading/release performance and cytotoxicity of the formulation.

#### 3.2.3. Effect of Centrifugation Parameters

Different centrifugation parameters were varied to investigate their effect on the properties of DSPC:Chol:PEG40s liposomes produced using the RIAC (at an anti-solvent temperature of 60 °C). These included the relative centrifugal force (rcf) and centrifugation time. As shown in Figure 6A, there was only a slight tendency for liposome size and the PDI to decrease with increasing the centrifugal force, and this effect was not statistically significant. It is hypothesised that increasing the centrifugal force might influence liposome production in two distinct ways. Firstly, it increases the flow velocity through the spiral mixing channel, leading to enhanced advection-dominated mixing. This is confirmed in studies utilising flow-through reactors with similar channel architectures, showing that increased flow rate levels can enhance and accelerate mixing between a lipid organic solution and an aqueous phase [57]. Specifically, it has been reported that vortical secondary flows (also known as Dean vortices) can significantly increase mixing efficiency between chemical species within curved channels and that this effect increases at greater fluid velocities [58]. The resulting increase in the efficiency and rapidity of mixing between the solvent and anti-solvent within the RIAC would thus lead to the self-assembly of liposomes having a smaller size and narrower size distribution.

Secondly, increasing the centrifugal force might also have the undesired effect of causing disruption and/or aggregation of liposomes due to the increased shear force. This may overall result in an increase in liposome size and size dispersity, as observed in a previous study where PC liposomes were produced using the RIAC [25].

This latter effect, however, was not observed to the same extent in the present study, likely due to the presence of PEG40s that mitigated liposome aggregation. Consequently, the effect of increased mixing efficiency likely compensated for the increased propensity to disrupt and/or aggregate liposomes, overall resulting in only a marginal (not statistically significant) reduction in liposome size and size dispersity when the centrifugal force was increased.

The other factor that may explain the differences between the present and previous studies is the variation in lipid formulations employed. For example, Yanar et al. [33] investigated the effect of flow rate on PC-based and DPPC-based liposomes produced with a millimetre-scale reactor and having comparable channel dimensions and architecture to the ones used in the present study. They observed contrasting effects of total flow rate on liposome dimensions, depending on the lipid formulation used.

As illustrated in Figure 6B, increasing the centrifugation time from 3 min to 4 min appeared to increase the liposome size while decreasing the PDI (from 0.17 to 0.10); however, as the centrifugation time was further prolonged from 4 min to 10 min, the liposome size slightly decreased while the PDI significantly increased (from 0.10 to 0.24), and both these changes were statistically significant.

It should be noted that from the initial testing on the RIAC, each reservoir containing 2 mL of liquid could be fully emptied within 2.5 min when operated at 2000 rcf. Therefore, extending the centrifugation time beyond 2.5 min had the potential effect of inducing liposome sedimentation at the bottom of the centrifuge tube and exposing the formed liposome pellet to shear flow for prolonged periods of time. Shear forces induced by centrifugation might have beneficial effects, i.e., inducing homogenisation of liposomes, as investigated by Ulrich et al. [59]. However, when the centrifugation time was increased beyond 4 min, potential liposome disruption and aggregation may have caused the observed increase in liposome size and size dispersity.

Overall, as liposomes produced after 4 min of centrifugation at 2000 rcf displayed a therapeutically relevant size (174 ± 6.58 nm) with the lowest size dispersity (PDI = 0.10 ± 0.04), these production parameters were chosen for subsequent experiments. It is, however, important to clarify that these experimental conditions and parameter values were not identified through a systematic optimisation process. Systematic methods, such as Design of Experiments (DOE) algorithms, could be employed in the future to identify optimal operating conditions for the RIAC that would produce liposomal dispersions of the desired properties.

#### 3.2.4. Effect of Lipid Concentration

The effect of the initial lipid concentration on liposome size was investigated by varying the total lipid concentration whilst maintaining a constant liposomal formulation of DSPC:Chol:PEG40s. As shown in Figure 7, liposomes produced with a total lipid concentration above 10 mM (DSPC:Chol:PEG40s > 5:5:0.5 mM) tended to have a greater size than those produced at <10 mM. The lipid concentration had a significant effect on both the size and PDI of the produced liposomes (*p* value < 0.0001). When a greater lipid concentration is employed, the increased number density of lipid molecules available for self-assembly might result in larger lipid bilayer fragments, which will in turn have an increased tendency to collide and assemble into larger liposomes. Interestingly, this increase in liposome size was not accompanied by increased size dispersity, which could be attributed to the stabilising action of PEG.

Previous studies using millimetre-scale flow reactors of comparable architecture demonstrated a correlation between lipid concentration and liposome size. For instance, Yanar et al. [33] showed a liposome size increase from 125 nm to 245 nm when the lipid concentration was increased from 5 to 200 mM, and a previous study using the RIAC also showed a liposome size increase from 130 to 300 nm when the lipid concentration was increased from 20 to 80 mM [25]. These changes were not observed to a comparable extent in the present study, likely due to the total lipid concentration range (2 to 20 mM) being smaller than the ones evaluated in previous research. Future investigations could therefore assess a wider lipid concentration range, which would also potentially enable greater liposome production rates.

Overall, the findings from the present study demonstrate that DSPC-based liposomes of comparable size and with a relatively low size dispersity can be produced using the RIAC over a range of total lipid concentrations between 2 and 20 mM.

### 3.3. Production of Functionalised Liposomes Using the RIAC

To explore whether the developed production method could be applied to more complex formulations, two types of functionalised liposomes were produced. The experimental conditions employed corresponded to the ones that individually led to the most favourable liposome properties (i.e., centrifugation time: 4 min, centrifugal force: 2000 rcf, and anti-solvent temperature: 60 °C).

With the development of gene therapy and mRNA-based vaccines, cationic liposomes have become a highly attractive drug delivery system due to their biocompatibility, efficient transfection performance, and ability to load nucleic acids via electrostatic interaction [60]. Therefore, the cationic liposome was chosen as one of the functionalised liposomal formulations to be produced using the RIAC. The design of this cationic formulation was based on the 1:1 ratio of cationic lipids (EPC) to neutral lipids (DSPC) as well as the 1:1 ratio of cholesterol to other lipids, as adopted frequently in DOTAP:chol liposome formulations [61]. The other functionalised formulation investigated was the biotinylated liposome, in which a biotin moiety was introduced to endow the liposome with a targeting ability and facilitate cellular internalisation [62]. Considering the important effect of PEG on liposome properties, as revealed in previous experiments, a half molar amount of PEG40s was replaced with DSPE-PEG-biotin to maintain a constant total amount of PEGylated moieties across formulations. As shown in Table 2, the produced functionalised liposomes had a therapeutically desirable size (120.1 nm for cationic and 118.9 nm for biotinylated liposomes) with low size dispersity (~0.17 for both formulations). The positive zeta potential value (i.e., a measure of electrostatic charge) of 81.43 mV for cationic liposomes further corroborated the successful incorporation of EPC within the liposome membrane bilayer.

Overall, the results demonstrate the ability of the RIAC to produce liposomal formulations containing functional moieties, which could be evaluated in the future for potential application in targeted drug delivery and the transport of genetic material. This would require further evaluation of drug loading and its effect on liposome size and stability, as well as liposome pharmacokinetic behaviour and biological compatibility. For example, previous work utilising comparable flow-reactor architectures and lipid formulations has shown that the loading of either hydrophilic or hydrophobic drugs causes a reduction in liposome mean diameter [32].

### 3.4. Super-Resolution Imaging of Liposomes Produced Using the RIAC

A number of liposome-sizing technologies have been developed over the years; however, they all suffer from some limitations that hinder their ability to measure the ‘true’ liposome size. This has driven research on alternative liposome-sizing and characterisation approaches. For example, Ingebrigtsen and Brandl showed that size exclusion chromatography (SEC) coupled with photon correlation spectroscopic particle size analysis led to better results than photon correlation spectroscopy (PCS) alone [48]. However, the complex setup and measurement uncertainty for a yield of large particle size differences still pose major drawbacks. To obtain incremental information on the liposome dimensional properties, a dSTORM super-resolution imaging protocol was implemented in the present study. As illustrated in Figure 8A, dispersed liposomes could be distinguished and imaged using the developed super-resolution protocol. Previous studies have demonstrated the applicability of super-resolution microscopy to investigate sub-cellular structures such as actin filaments, nanocarriers [49], lipid bilayer phases [50], and domains [51,52]. To the best of our knowledge, this is the first time that a STORM-based technique has been successfully applied to image, size, and qualitatively characterise nanoscale liposomes. The high-throughput feature of STORM implemented with a custom protocol, facilitated the ultimate characterisation of the produced liposomes. As reported in Figure 8B, the CODI protocol was used for the classification and analysis of more than 3000 clusters. Compared to DLS data, the super-resolution method offered further insights into particle dimensions that are based on direct imaging. From the quantitative analysis reported in Figure 8C, the radius of gyration for the liposomes was found to average out at 90 nm. The overall distribution was found to result in liposomes concentrated at a radius lower than 200 nm. Taken together, these observations provide direct qualitative evidence of the produced liposomal dispersion, combined with accurate quantitative determination of its size distribution. They also provide confirmation that the liposomal dispersion achieved using the RIAC has a size distribution that is compatible with therapeutic (i.e., drug delivery) applications.

## 4. Conclusions

The present study reports on the production of liposomes comprising constituents that are common to therapeutic formulations (such as DSPC, cholesterol, and PEGylated moieties), using a simple and cost-effective production method based on a centrifuge-actuated reactor (known as reactor-in-a-centrifuge, or RIAC). This device was employed in previous research to produce both organic and inorganic particulate systems, including silver nanoparticles and quercetin nanocrystals. The RIAC can be manufactured using a benchtop 3D printer within 8 h, at a material cost as low as GBP 1.2 per device (at the time of writing). Notably, the technical drawings and 3D printing specifications for the device are freely available online for adoption and modification across different laboratories. Liposome production could be achieved by actuating the device using a centrifuge operated at centrifugal forces between 1000 and 2000 rcf, which are compatible with standard and widely accessible laboratory centrifuges. The device therefore enables flow synthesis without cumbersome syringe pumps, time-consuming experimental steps, or reliance on specialist expertise (as for traditional microfluidic-based methods). Each individual device can achieve a liposome production rate > 8 mg/mL, and the overall throughput could be increased by simultaneously operating multiple devices within a single centrifuge rotor.

The protocol of DSPC:cholesterol liposome production was progressively improved by individually varying different formulation- and production-related parameters. Overall, the improved production method was capable of generating liposomes with a therapeutically relevant size (<200 nm) and low size dispersity (PDI < 0.2).

To the best of our knowledge, this is the first report demonstrating the production of DSPC- and cholesterol-based liposomes, with dimensional properties suitable for clinical use, using a centrifuge-actuated flow-through method. The utility of this method in producing liposomes with therapeutic potential was further corroborated by the successful production of two functionalised formulations, incorporating a charged lipid and a biotinylated moiety, respectively. Furthermore, it was demonstrated for the first time that super-resolution dSTORM imaging coupled with quantitative image analysis can be employed as a viable, non-destructive technique to determine liposome size distribution and evaluate particle morphology.

In conclusion, the flow-through liposome production method proposed in this study has the potential to become an effective and versatile approach to simplify the synthesis of therapeutic liposomal formulations, due to the single-step and pump-free nature of the process and its applicability within non-specialist laboratory settings. Future studies could investigate scaling-up strategies to achieve greater production rates, the incorporation of additional features or ‘units’ within the device for the loading of model drugs or biologically active compounds (such as genetic materials), and the evaluation of liposome biocompatibility.

## Figures and Tables

**Figure 1 micromachines-14-01763-f001:**
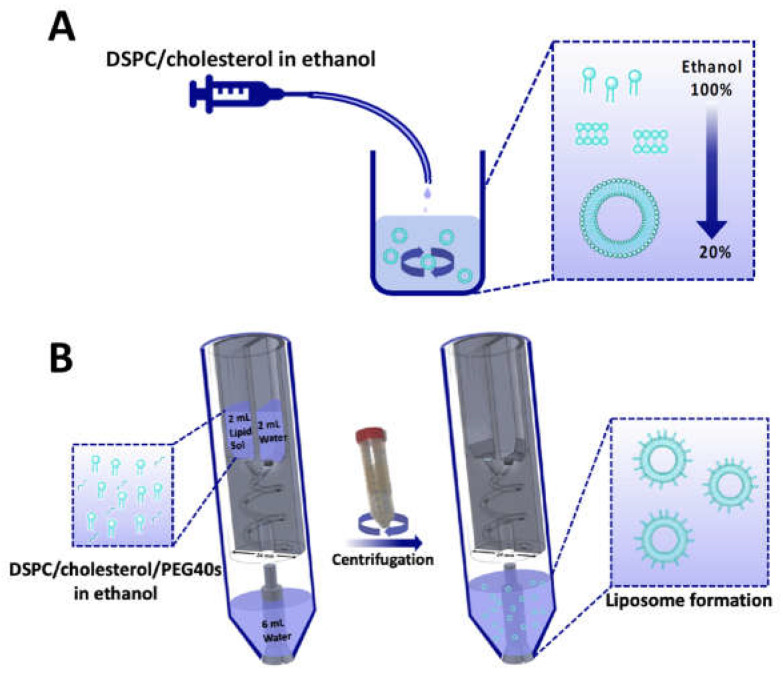
Schematic illustration of the laboratory-scale liposome production methods employed in the present study, which include (**A**) ethanol injection and (**B**) reactor-in-a-centrifuge (RIAC). Both methods rely on the so called ‘solvent exchange mechanism’ to induce liposome formation. In the RIAC, the channel architecture is manufactured within a cylindrical structure with diameter of 24 mm. Figure modified from Cristaldi et al. [25].

**Figure 2 micromachines-14-01763-f002:**
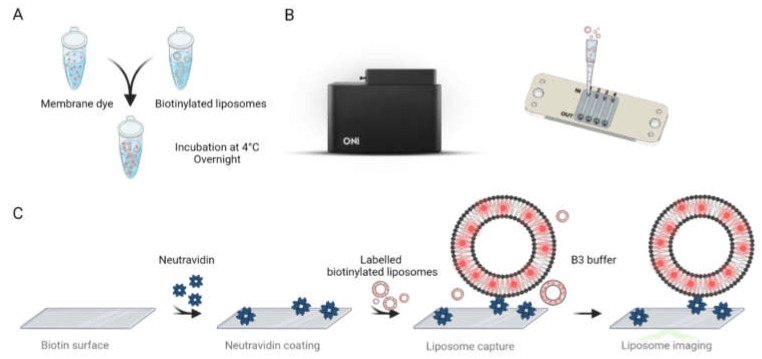
(**A**) Labelling protocol for biotinylated liposomes. (**B**) Nanoimager (**Left**) and schematic representation of ONI’s imaging chip with labelled inlets, outlets, and flow lanes (from 1 to 4) (**Right**). (**C**) Schematical workflow for specifically capturing and imaging the labelled liposomes.

**Figure 3 micromachines-14-01763-f003:**
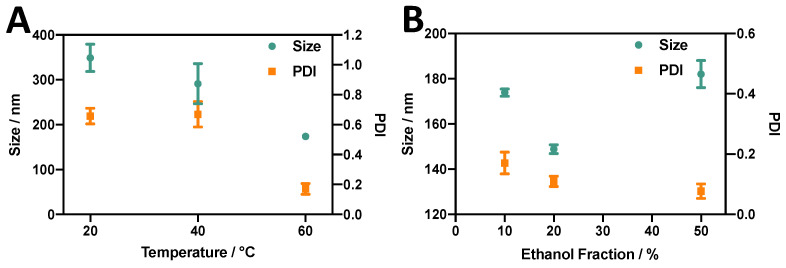
Effects of (**A**) anti-solvent temperature and (**B**) % ethanol fraction (by volume) on the size and PDI of DSPC:chol liposomes produced via ethanol injection.

**Figure 4 micromachines-14-01763-f004:**
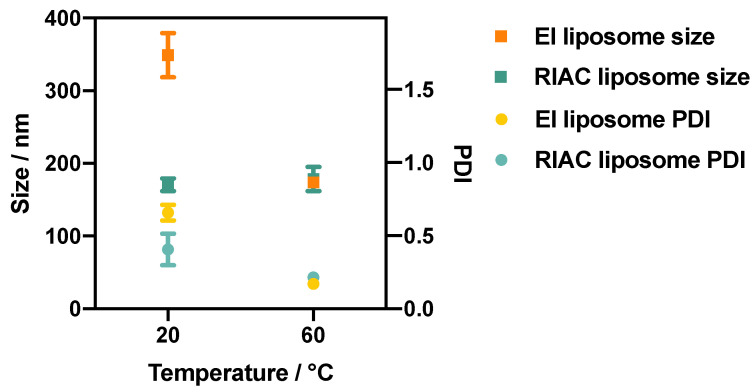
Size (squares) and PDI (circles) of DSPC:chol (1:1) liposomes produced via ethanol injection (EI) and using the RIAC, at anti-solvent temperatures of 20 °C and 60 °C. The ethanol fraction in these experiments was equal to 20% (by volume).

**Figure 5 micromachines-14-01763-f005:**
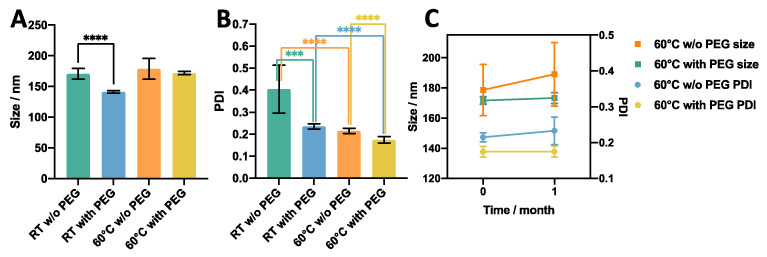
Effect of the presence of a PEG moiety on (**A**) size, (**B**) size dispersity (PDI), and (**C**) stability of liposomes produced using RIAC. The liposome formulation was DSPC:Chol:PEG40s (molar ratio 10:10:1). *** *p* < 0.001, **** *p* < 0.0001.

**Figure 6 micromachines-14-01763-f006:**
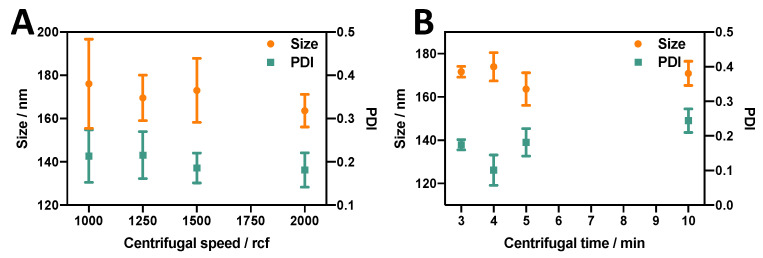
Effect of (**A**) centrifugal force (rcf) and (**B**) centrifugation time on the size and PDI of liposomes produced using the RIAC. The formulation employed was DSPC:Chol:PEG40s (10:10:1), and liposomes were produced at the anti-solvent temperature of 60 °C.

**Figure 7 micromachines-14-01763-f007:**
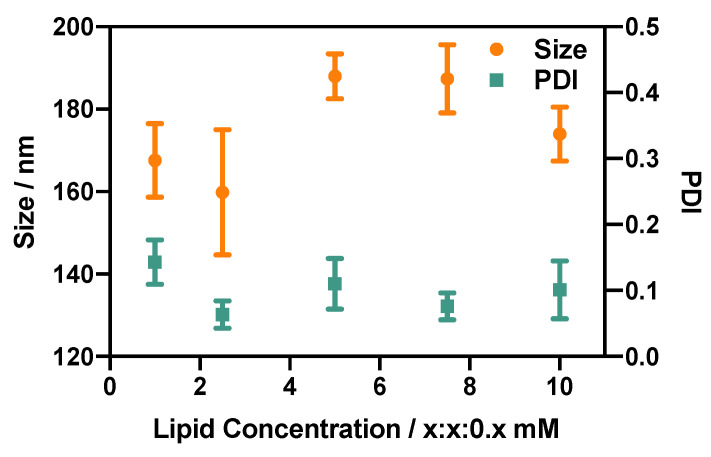
Effect of lipid concentration on the size and size dispersity (PDI) of DSPC:Chol:PEG40s liposomes. *x* refers to the concentration (in mM) of DSPC and cholesterol, while the concentration of PEG40s is one-tenth of *x*. Liposomes were produced using the RIAC (4 min centrifugation at 2000 rcf) at an anti-solvent temperature of 60 °C.

**Figure 8 micromachines-14-01763-f008:**
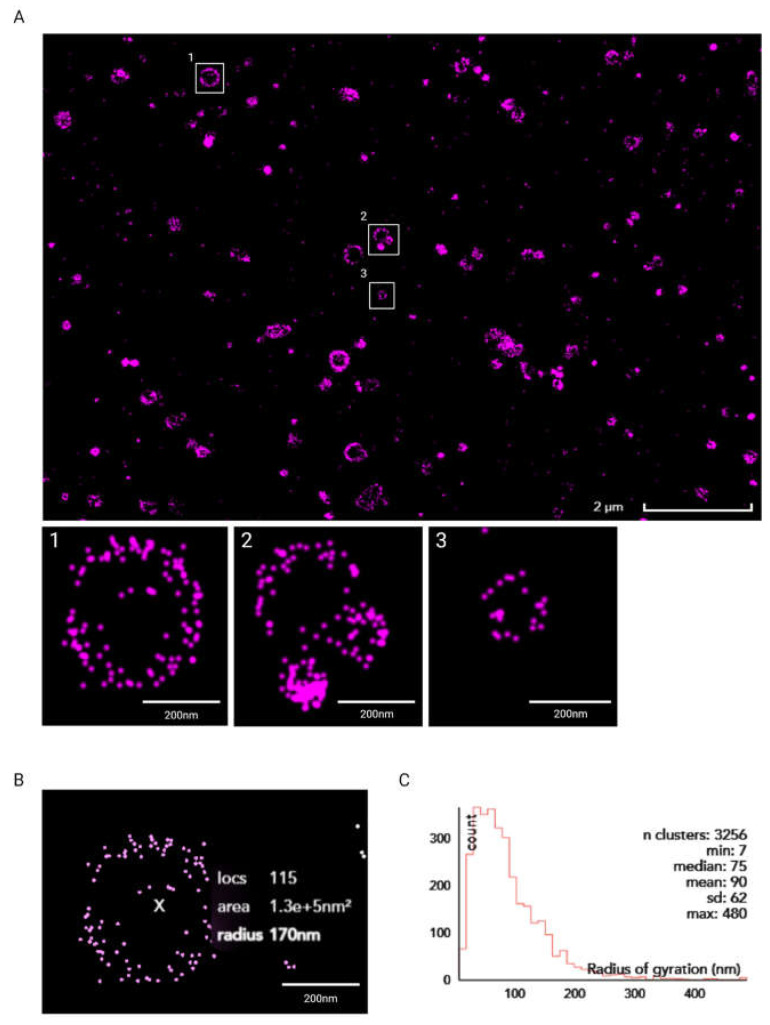
(**A**) Super-resolution dSTORM image of liposomes obtained using the Nanoimager S and further processed using CODI. Scale bar: 2 µm (main image) and 200 nm (insets 1–3, showing zoomed-in views of selected liposomes). (**B**) Clustering analysis of liposome in inset 1 using CODI. Scale bar: 200 nm. (**C**) Histogram plot of liposomes’ radius of gyration, in nm.

**Table 1 micromachines-14-01763-t001:** List of liposomal formulations investigated in the present study and the corresponding production method for each formulation.

Liposome Production Method	Liposome Formulations Evaluated
Ethanol injection	(1) DSPC:chol
Reactor-in-a-centrifuge (RIAC)	(2) DSPC:chol
	(3) DSPC:chol:PEG40s
	(4) DSPC:chol:EPC:PEG40s
	(5) DSPC:chol:PEG40s:DSPE-PEG-biotin

**Table 2 micromachines-14-01763-t002:** Composition (with corresponding molar concentrations), size (in nm), PDI, zeta potential (in mV), and concentration of two types of functional liposomes (cationic and biotinylated) produced using the RIAC.

Liposome Properties	Cationic Liposome	Biotinylated Liposome
Composition (mM)	DSPC:chol:EPC:PEG40s (1:2:1:0.2) 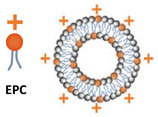	DSPC:chol:PEG40s:DSPE-PEG-biotin (2:2:0.1:0.1) 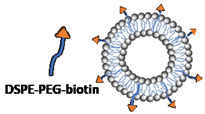
Size (nm)	120.1 ± 0.6	118.9 ± 1.6
PDI	0.17 ± 0.01	0.17 ± 0.02
Zeta potential (mV)	81.43 ± 4.48	NA
Concentration	9.62 × 10^9^ ± 5.09 × 10^8^	2.12 × 10^10^ ± 4.04 × 10^9^

## Data Availability

The data presented in this study are available on request from the corresponding author.

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
