# Peer review of "Rapid Production of Nanoscale Liposomes Using a 3D-Printed Reactor-In-A-Centrifuge: Formulation, Characterisation, and Super-Resolution Imaging"

_micromachines, 2023, doi:10.3390/mi14091763_

Round 1

Reviewer 1 Report

In this work, Authors showed the formulation of liposomes using an already published microfluidic device. In the previous publication, liposomes were produced as well, with another excipient. Even if the effort of the Authors is good, I believe that the manuscript need an upgrade before to be considered for the publication. In particular my comments are the following:

-Line 40: The marketed vaccines rely on lipid nanoparticles that are different from liposomes; Please consider this;

-line 48: Commercially available liposomes are not produced by TLE, and the other techniques that usually are more common for lab-scale production. Please rephrase;

-line 72: The production ratio is not that low (few mg/mL). Even with commercially available systems it is possible nowadays to produce with high rate. Please correct.

-line 93-94: Here ethanol injection is introduced. I suggest to move this part where the batch methods are described.

-Figure 1: please modify the A part because it is not clear that the water is in the becker 

-Ref 21, 22 in line 108 don't seem to be appropriate

-line 118: please rephrase because it is written "more relevant" but respect to what? 

-materials 2.1: It is not clear how the solvent has been removed from the formulation

-materials 2.2 (correct the numeration): specify the 3DP technology used

-materials 2.3: report the formulations considered in a table to be more clear to the readers

-line 177: what does it mean "without compromising concentration"? When a dilution happen the concentration obviously change so this statement seems not clear.

-line 180: how it is possible to mantain the temperature during the extrusion process? A pre-heating cannot be considered as the production temp.

-line 183: A single set of microfluidic parameters cannot be optimized for all the formulations. Each time an excipient change, the production parameters should be optimized again.

- In the discussion some of the results presented are not statistically significant so it would be better to reconsider the discussion to be less speculative on the results.

-Please add statistics to all the figures

-line 415: Have higher concentrations been tested? 

-line 444: This statement is speculative since it has not been proved that genetic material is efficiently loaded into the produced liposomes. This is an aspect that the authors should take in account to increase the quality of the paper. Even a model genetic material can be selected but its loading should be proved and quantified.

-line 480: therapeutically relevant based on what? In this work there are no biobased studies. Even a simple cell compatibility could be useful to improve the quality.

-Conclusions should be more coincise without repeating the results. It happen from line 490 to 510 more or less. Please rewrite the conclusions.

Please check for few typing errors.

Author Response

See file attached.

Reviewer 2 Report

The manuscript reports the fabrication of medicated liposomes using 3DP. Several parameters were investigated such as the concentration of polyethylene glycol, temperature, centrifugal time and speed, and lipid concentration about their influences on the size of liposomes. These contents are interesting and fall within the scope of Micromachines. It can be accepted for publication after minor revision.

1) The arrangements of Figire 3, 5, 6 should be within the scope of texts.

2) It should be better to include a model drug (particularly one of the poorly water-solube drugs) with the lipid, at least some discussion about the influence of loading drug on the formation can be added.

3) The quality of Figure 8 can be improved.

4) Almost no references are publications within the most recent three years.  

Some small places can be improved, e.g. blank between value and unit in Figures.

Round 2

Reviewer 1 Report

The Authors have addressed most of the comments improving the overall work in a good manner. I believe it is now suitable for publication in the Journal

Please check minor typing and spelling errors.